immunology/microbiology

Apis cerana, gut microbiota, antimicrobial peptide, immune system, Nosema ceranae

**Author for correspondence:**
Fuliang Hu
e-mail: flhu@zju.edu.cn

# Apis cerana gut microbiota contribute to host health though stimulating host immune system and strengthening host resistance to Nosema ceranae

Yuqi Wu, Yufei Zheng, Yanan Chen, Gongwen Chen, Huoqing Zheng and Fuliang Hu

College of Animal Sciences, Zhejiang University, Hangzhou, People's Republic of China

YW, 0000-0003-3301-3927

Gut microbial communities play vital roles in the modulation of many insects' immunity, including *Apis mellifera*. However, little is known about the interaction of *Apis cerana* gut bacteria and *A. cerana* immune system. Here in this study, we conducted a comparison between germ-free gut microbiota deficient (GD) workers and conventional gut community (CV) workers, to reveal the possible impact of gut microbiota on the expression of *A. cerana* antimicrobial peptides and immune regulate pathways. We also test whether *A. cerana* gut microbiota can strengthen host resistance to *Nosema ceranae*. We find that the expression of *apidaecin*, *abaecin* and *hymenoptaecin* were significantly upregulated with the presence of gut bacteria, and JNK pathway was activated; in the meanwhile, the existence of gut bacteria inhibited the proliferation of *Nosema ceranae*. These demonstrated the essential role of *A. cerana* gut microbiota to host health and provided critical insight into the honeybee host–microbiome interaction.

## 1. Introduction

In animal gastrointestinal tracts live complex assemblages of microorganisms, some of which are emerging as key players in governing host health [1]. In particular, the gut bacteria communities of insects have been found involved in food digestion, nutrient provisioning, host development and intraspecific

communication, and they generally can contribute to host health through immune system stimulation as well as conferring resistance against pathogens [2–4].

The honeybee (*Apis* spp.) gut is colonized by 8 to 10 phylotypes of bacteria, namely, *Snodgrassella alvi*, *Gilliamella apicola*, *Lactobacillus* spp., *Bifidobacterium* spp., *Frischella perrara*, *Bartonella apis*, *Parasaccharibacter apium* and *Commensalibacter* spp. [5]. Due to their social behaviour and the ways of acquiring gut microbiota, honeybees consistently harbour a highly conserved gut community, many of which have coevolved with their host for millions of years [6]. Thus, it is not surprising that these gut bacteria are tightly intertwined with the physiology of honeybees. Study has demonstrated that *Apis mellifera* gut microbiota possesses a large repertoire of metabolic capabilities [7]. For example, Zheng *et al.* [8,9] have shown that *A. mellifera* gut microbiota can promote host development and participate in the metabolism of toxic sugars.

Several studies also clearly demonstrated that *A. mellifera* gut microbiota make a strong contribution to their hosts' health. One way they can do this is by promoting the expression of *apidaecin* and *hymenoptaecin* [10], or strongly activating the host immune system as caused by the gut bacteria *S. alvi* and *F. perrara* [11]. Further, a recent study of antibiotic-treated *A. mellifera* revealed that destruction of their gut bacteria could increase the vulnerability of honeybee to *Nosema* infection [12]. *Nosema ceranae* is a microsporidia parasite originally found in the eastern honeybee (*Apis cerana*) in 1996 [13]. Now it is a common pathogen infecting European honeybee species and it is highly pathogenic to its novel host [14]. *Nosema ceranae* infection impairs midgut integrity and alters the energy demand in *A. mellifera*, and it can also significantly suppress bee immune response and modify pheromone production in *A. mellifera* workers and their queens, leading to precocious foraging [15].

However, most studies of honeybee host–microbiome interactions have focused on *A. mellifera*, leaving our knowledge about the function of *A. cerana* gut symbionts quite limited. The *A. cerana* honeybee's native range spans southern, southeastern and eastern Asia. This species is in the same subgenus as the western honeybee (*A. mellifera*), which is naturally spread though Africa and Europe. Both species share many biological similarities, in terms of their physiology, behaviour and life history, and their guts are both colonized by the same phylotypes of bacteria. Interestingly, studies have revealed a distinctive strain-level diversity between these two species [6,16], which could be a source of functional diversity [17]. Therefore, research on the interaction between *A. cerana* and its gut microbiota could not only shed light upon the functioning of *A. cerana* gut microbiota functions but also provide insight into the functional differences between *A. cerana* and *A. mellifera* gut microbial communities.

In the current study, we focused on the contribution of *A. cerana* gut microbiota to host health in two key aspects: (i) the interaction between gut microbiota and immune system of *A. cerana*, and (ii) the interaction between gut microbiota and parasites of *A. cerana*. By comparing workers colonized with or without gut bacteria, we compared the gene expression of both antimicrobial peptides (AMPs) and the regulators of immune pathways, and we also evaluated the role of gut microbiota in *Nocema* resistance. We found that *A. cerana* gut microbiota can upregulate the expression of genes *apidaecin*, *abaecin*, *hymenoptaecin* and *vitellogenin*, regulate key components of the JAK/STAT and JNK pathways and demonstrate their contribution to host resistance against *N. ceranae*.

# 2. Material and methods

## 2.1. Rearing of honeybees

*Apis cerana* colonies for the experiments and *A. mellifera* colony for *Nosema* spore were all kept in the apiary maintained at the Honey Bee Research Laboratory in the College of Animal Sciences, Zhejiang University, Hangzhou, China. Experiments were replicated three times using three different *A. cerana* colonies, and in each replicate, *A. cerana* workers and *A. cerana* gut bacteria from the same colony were used.

Gut microbiota deficient (GD) *A. cerana* workers and conventional gut community (CV) *A. cerana* workers were obtained using the protocol described by Zheng *et al.* [8]. Briefly, late stage pupae (dark-eyed) were removed from brood frames and transferred into sterile 48-well cell culture plates. These plates were placed in an incubator at $34°C \pm 1°C$ with $80\% \pm 5\%$ relative humidity until bees emerged. Workers emerged within the 48 h after transformation were collected for the experiments.

The newly emerged germ-free bees were randomly assigned to GD or CV groups (50 workers per group). GD workers were supplied with sterile pollen and sterile sugar water (50% sucrose solution, w/v), while CV workers were supplied with food containing gut bacteria for 5 days and then switched to sterile pollen and sterile sugar water. For the quantification of bacterial loads of GD and

CV bees, six workers from each cage were sampled on day 5 after emergence. In addition, we have also sampled 10 forager workers from the field to serve as a positive control. For the gene expression profiling, six workers per cage were collected from each of the GD and CV groups on day 14. Experiments were replicated three times, with a total of three cages used for GD and three cages for CV.

To analyse the interaction between *A. cerana* gut bacteria and *Nosema*, newly emerged germ-free workers were randomly assigned to four different groups (50 bees per group). GDT: germ-free workers were inoculated individually with 2 µl of a syrup-spore suspension containing $1 \times 10^5$ *Nosema* spores [18] and supplied with sterile pollen and sterile sugar water; GDC: germ-free workers were inoculated with PBS and supplied with sterile pollen and sterile sugar water; CVT: germ-free workers were inoculated individually with 2 µl syrup-spore suspension containing $1 \times 10^5$ *Nosema* spores and colonized with gut homogenates; CVC: germ-free workers were inoculated with PBS and colonized with gut homogenates. The number of dead bees were recorded daily and removed. Six live bees from each cage were randomly sampled to monitor *Nosema* spore numbers on 7th and 14th day post infection (dpi), respectively. Experiments were replicated three times, with a total of 12 cages: three cages each for GDT, GDC, CVT and CVC treatment groups.

## 2.2. Preparation of *Apis cerana* gut bacteria

Ten *A. cerana* forager workers were sampled from the entrance of bee hives; their hindguts were dissected immediately and homogenized together in 1 ml PBS. This gut homogenate was centrifuged at $10\,000g$ for 10 min and the supernatant were removed to eliminate the possible viral contamination. Then 1 ml PBS was used to resuspend the bacteria, from which a 100 µl suspension was added to and mixed with sterilized pollen.

## 2.3. Quantification of bacterial loads in the gut of honeybees

Bacterial loads of GD and CV workers were determined by qPCR [19], using the universal bacterial 16S rRNA primers as listed in table 1. Workers were immobilized at 4°C and their whole guts were immediately dissected. From these, their DNA was extracted using the TIANamp Stool DNA Kit (Tiangen Biotech Co., Ltd, Beijing, China) according to the manufacturer's protocol and then used for the qPCR. The absolute quantification of 16S rRNA copy numbers was quantified using the StepOne Plus real-time PCR system for which the thermal cycling conditions were as follow: initial denaturing step of 95°C for 30 s, 40 amplification cycles of 95°C for 5 s and 60°C annealing for 30 s, with a melt curve analysis done from 60°C to 95°C at 0.5°C/5 s increments to confirm expected dissociation curves. The qPCR reaction mixtures were set up with 1 µl of DNA, 0.2 µl of forward and reverse primers (10 µM), 5 µl of TB Green™ *Premix Ex Taq* (Takara Biomedical Technology Co., Ltd, Beijing, China) and 3.6 µl of distilled water. All DNA samples were replicated in three wells.

## 2.4. Profiling the gene expression levels of antimicrobial peptides and immune pathways

Workers were immobilized at 4°C and their whole abdomens were immediately dissected and used for RNA extractions. We assayed the transcript levels of the following genes: *apidaecin, abaecin, defensin 1, defensin 2, hymenoptaecin, relish, dorsal, basket, Imd, toll, domeless, kayak, hopscotch* and *vitellogenin (Vg)*, with the housekeeping gene *actin related protein 1 (Arp1)* chosen as the reference control. The primers used are listed in table 1.

Total RNA was extracted with the RNApure Total RNA Kit (Aidlab Biotechnologies Co. Ltd, Beijing, China) according to the manufacturer's protocol. The cDNA synthesis reaction was performed using 0.5 µg of total RNA with the PrimeScript™ RT Master Mix (Takara Biomedical Technology Co., Ltd, Beijing, China). The concentration and quality of this RNA was determined using the Nanodrop 2000 (Thermo Fisher Scientific, MA, US). The qPCR reaction mixtures were set up with 1 µl of cDNA (10X diluted) as described above. The relative expression levels of target genes were calculated by applying the $2^{-\Delta\Delta Ct}$ method [22].

## 2.5. Isolation and counting of *Nosema* spore

*Nosema* spores for inoculation were freshly isolated from a heavily infected *A. mellifera* colony kept at the experimental apiary; hence, all the *N. ceranae* spores used in this study were isolated from a single colony. A total of 30 midguts were homogenized in 30 ml of distilled water. To sediment the *Nosema* spores, this

**Table 1.** The primers used in this study.

| target gene name and accession no. | sequence (5' to 3') | | gene classification | amplicon size (bp) | reference |
|---|---|---|---|---|---|
| *universal bacterial* 16S rRNA | F: | AGAGTTTGATCCTGGCTCAG | bacteria quantification | 328 | [19] |
| | R: | CTGCTGCCTCCCGTAGGAGT | | | |
| *Arp1* (XM_017059067.2) | F: | CTCACAGTGTTCGCAACTCG | housekeeping gene | 206 | this study |
| | R: | CGAAACCGGCTTTGCACATA | | | |
| *abaecin* (XM_017063755.2) | F: | ATCTTCGCACTACTCGCCAC | antimicrobial peptide | 103 | this study |
| | R: | CCTGACCAGGAAACGTTGGA | | | |
| *apidaecin* (XM_017060818.2) | F: | CCAGATCCGCCTACTCAACC | antimicrobial peptide | 131 | this study |
| | R: | GGTTTAGCTTCACGGCGTAG | | | |
| *defensin 1* (XM_017050425.2) | F: | AGCCACTTGAGCATCCTGAG | antimicrobial peptide | 151 | this study |
| | R: | CCGTTCTTGCAATGACCTCC | | | |
| *defensin 2* (XM_017060723.2) | F: | TTTCGCGATTCTCGTCGCTA | antimicrobial peptide | 156 | this study |
| | R: | TGTCGTAGCAGTAGCGGTTC | | | |
| *hymenoptaecin* (XM_017049926.1) | F: | CGTGTTGGTTGTCTTCTGCG | antimicrobial peptide | 209 | this study |
| | R: | CACCATAGGCATCTCCCGTC | | | |
| *Vg* NM_001328484.1) | F: | ACCAACGACTTCATGGGACC | immune marker | 181 | this study |
| | R: | CGCTGTCGCTGATCACATTG | | | |
| *relish* (XM_017053040.2) | F: | TGAAGCTGGTGCATGTGTTG | IMD pathway | 105 | this study |
| | R: | CCTGCTTTTGCTGCAAGATGT | | | |
| *dorsal* (XM_017054012.2) | F: | TTTATCACGATTGTAGATGCTGC | *toll* pathway | 149 | *this* study |
| | R: | GGAGAAGTTGTTGCCATCGG | | | |
| *basket* (XM_028666319.1) | F: | AGGAGAACGTGGACATTTGG | JNK pathway | 243 | [20] |
| | R: | AATCCGATGGAAACAGAACG | | | |
| *Imd* (XM_017057615.2) | F: | TGTTAACGACCGATGCAAAA | IMD pathway | 153 | [20] |
| | R: | CATCGCTCTTTTCGGATGTT | | | |
| *toll* (XM_017053307.2) | F: | TCGATGTCCAACGGAGCAAA | toll pathway | 102 | this study |
| | R: | ACTTTCACAACGAAGGCCGA | | | |
| *domeless* (XM_028666710.1) | F: | TTGTGCTCCTGAAAATGCTG | JAK/STAT pathway | 180 | [20] |
| | R: | AACCTCCAAATCGCTCTGTG | | | |
| *kayak* (XM_028666620.1) | F: | CGACAGATCCGCAGAGAAAG | JNK pathway | 148 | [21] |
| | R: | CCTGTTGCAGCTGTTGTATC | | | |
| *hopscotch* (XM_028664411.1) | F: | ATTCATGGCATCGTGAACAA | JAK/STAT pathway | 141 | [21] |
| | R: | CTGTGGTGGAGTTGTTGGTG | | | |

homogenate was centrifuged at 3000 r.p.m. for 5 min and the supernatant was discarded. The centrifugation process was repeated twice to obtain a crude *Nosema* spore suspension. To obtain the pure *Nosema* spores, the suspension was then purified by the method of Percoll gradient centrifugation [14]. *Nosema* spore concentrations were determined by counting spore numbers within a haemocytometer chamber (Hausser Scientific 3100). *Nosema* spore concentration was calculated by the following equation: $((\text{total numbers of spores counted in the haemocytometer}) \times 4 \times 10^{6})/\text{square numbers}$. The spore-syrup suspension was freshly prepared by mixing the pure *Nosema* spores with 30% sucrose syrup before use. The species identity of *Nosema* spores was confirmed by PCR [23]. To count the *Nosema* spore numbers in individual worker, each bee's abdomen was smashed and macerated in 1 ml of distilled water. From this, a 10 µl tissue solution was placed in a haemocytometer to quantify *Nosema* spore numbers [24].

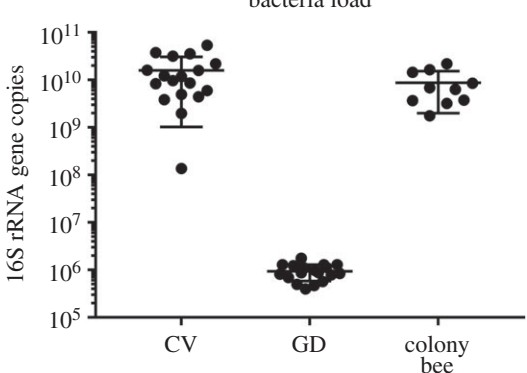

**Figure 1.** The bacterial colonization levels in the guts of GD worker ($n = 5$ bees $\times$ 3 replicates), CV workers ($n = 5$ bees $\times$ 3 replicates) and colony bees ($n = 10$). For each boxplot, the centre line displays the median, + indicates expression mean, the boxes correspond to the 25th and 75th percentiles and whiskers span the 10th–90th percentile.

## 2.6. Data analysis

Statistical analysis was carried out using SPSS software version 22.0. For the gene expression profiling and *Nosema* spore numbers, results from three replicates were pooled together and statistical significance was calculated using the independent sample *t*-test with Bonferroni-corrected *p*-values ($p = 0.05/3 = 0.016$). The Kaplan–Meier survival curve and Cox regression analyses were used for the survivorship data. All the figures were drawn in GraphPad Prism 7.

# 3. Results

## 3.1. *Apis cerana* gut microbiota stimulates host antimicrobial peptides and vitellogenin expression

The copy numbers of the 16S rRNA gene in the gut of *A. cerana* workers was significantly increased when inoculated with normal gut microbiota (Student's *t*-test, $t = 4.538$, d.f. $= 34$, $p < 0.001$), with a total of approximately $10^{10}$ bacterial cells found per gut of CV bees versus a total bacterial load of around $10^6$ cells per gut of GD bees (figure 1). Furthermore, the comparison between CV workers and colony workers revealed that they had similar bacteria loads in their gut (Student's *t*-test, $t = 1.438$ d.f. $= 26$, $p = 0.163$). These results verified that bees fed with homogenized guts were successfully colonized by honeybee gut bacteria.

We found that *A. cerana* gut microbiota had a significant impact on the expression of three AMP coding genes: *apidaecin* (Student's *t*-test, $t = 3.108$, d.f. $= 34$, $p = 0.004$), *abaecin* (Student's *t*-test, $t = 3.381$, d.f. $= 34$, $p = 0.002$) and *hymenoptaecin* (Student's *t*-test, $t = 3.065$, d.f. $= 34$, $p = 0.004$), which were increased 3.1-fold, 4.6-fold and 7.4-fold, respectively. The upregulation of *Vg* (Student's *t*-test, $t = 2.159$, d.f. $= 34$, $p = 0.015$), a common marker for overall honeybee health, was also detected in CV workers. By contrast, the transcripts for *defensin 1* (Student's *t*-test, $t = 0.015$, d.f. $= 34$, $p = 0.988$) and *defensin 2* (Student's *t*-test, $t = 1.696$, d.f. $= 34$, $p = 0.099$) did not show any significant differences between the CV and GD workers (figure 2).

## 3.2. *Apis cerana* gut microbiota actives JNK pathways

As shown in figure 3, gut bacteria colonization dramatically increased the expression of *basket* (Student's *t*-test, $t = 2.847$, d.f. $= 34$, $p = 0.012$) and *kayak* (Student's *t*-test, $t = 2.671$, d.f. $= 34$, $p = 0.012$) genes, which are major components of the JNK pathway. Interestingly, the expression levels of *domeless* (Student's *t*-test, $t = 2.309$, d.f. $= 34$, $p = 0.026$) and *hopscotch* (Student's *t*-test, $t = 2.073$, d.f. $= 34$, $p = 0.046$), both of which are immune regulators of JAK/STAT pathway, in the CV workers were not significantly higher than those of GD workers. Also, negligible interactions were found between *A. cerana* gut microbiota and toll as well as the IMD pathways, in that the expression level of key components of these two pathways, namely *relish* (Student's t-test, t $= 1.127$, d.f. $= 34$, $p = 0.304$), *dorsal* (Student's *t*-test, $t = 0.457$,

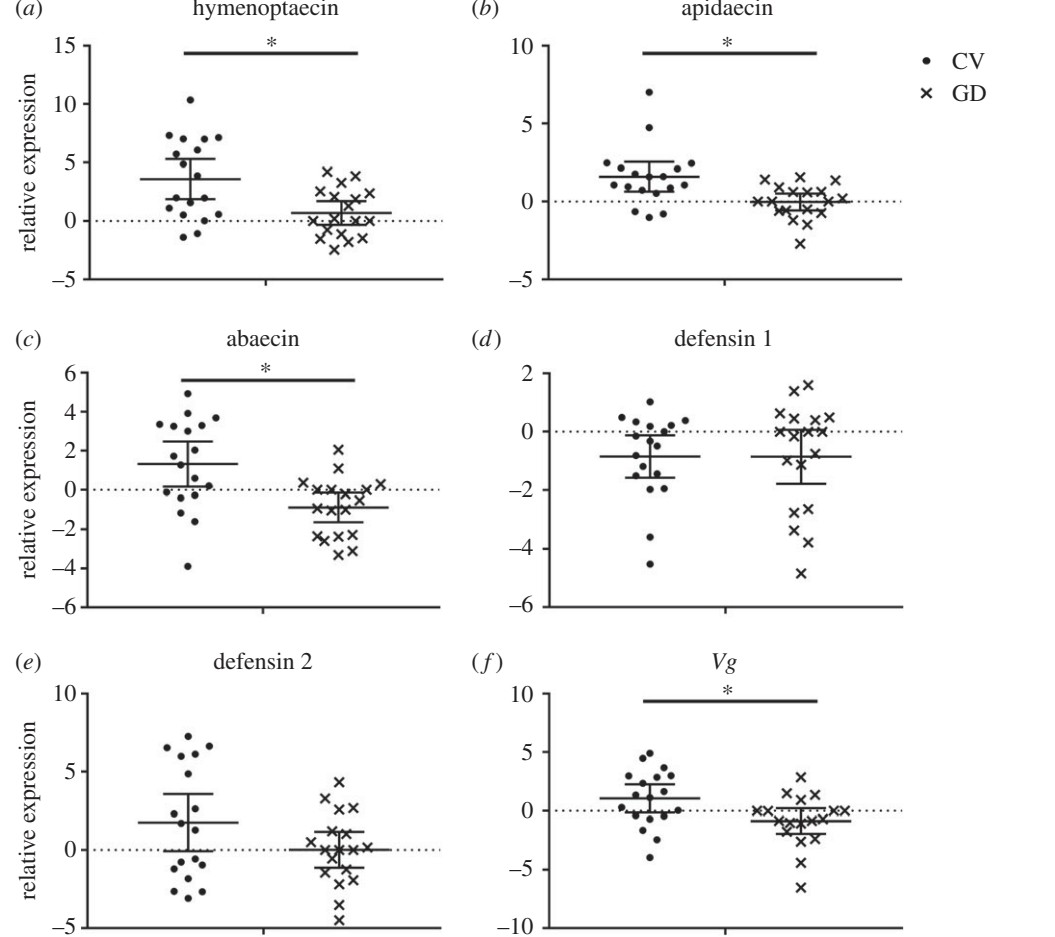

**Figure 2.** The expression level of AMPs and *Vg* in the abdomen of GD worker (*n* = 6 bees × 3 replicates) and CV workers (*n* = 6 bees × 3 replicates). The relative expression was log₂ transformed. Means ± 95 CI are shown by the black bars and whiskers. *represents the significant difference (*p* < 0.016).

d.f. = 34, *p* = 0.742), *toll* (Student's *t*-test, *t* = 0.286, d.f. = 34, *p* = 0.777) and *Imd* (Student's *t*-test, *t* = 0.076, d.f. = 34, *p* = 0.913), did not undergo significant regulation with gut bacteria present in the host (figure 3).

### 3.3. *Apis cerana* gut microbiota promote host resistance against *Nosema ceranae* infection

The numbers of *Nosema* spores in honeybee workers' gut were significantly affected by the presence of gut bacteria. An average of $2.73 \times 10^7$ spore per bee were counted in the gut of GDT workers, whereas only $1.23 \times 10^7$ spores were found in CVT workers at 14 dpi (Student's *t*-test, *t* = 6.561, d.f. = 34, *p* < 0.001); however, no significant difference were observed at 7 dpi (Student's *t*-test, *t* = 1.58, d.f. = 34, *p* = 0.123, figure 4*a*). Additionally, no spores were identified in the GDC and CVC bees, which demonstrated that workers used in the experiments were *Nosema* free.

We also noticed that although the spore numbers were significantly different between CVT and GDT workers, the survival rate of *Nosema*-treated workers did not differ greatly during the experiment period (figure 4*b*). Furthermore, no increase in mortality rates was observed in workers lacking their microbiota, which agrees with similar findings on *A. mellifera* [25].

## 4. Discussion

Many studies have shown that gut bacteria are key players in immune modulation and are essential for a healthy immune system [26,27]. In all stages of their life cycle, insects are threatened by a multitude of predators, parasites, parasitoids and pathogens. To counteract these threats, insects have evolved

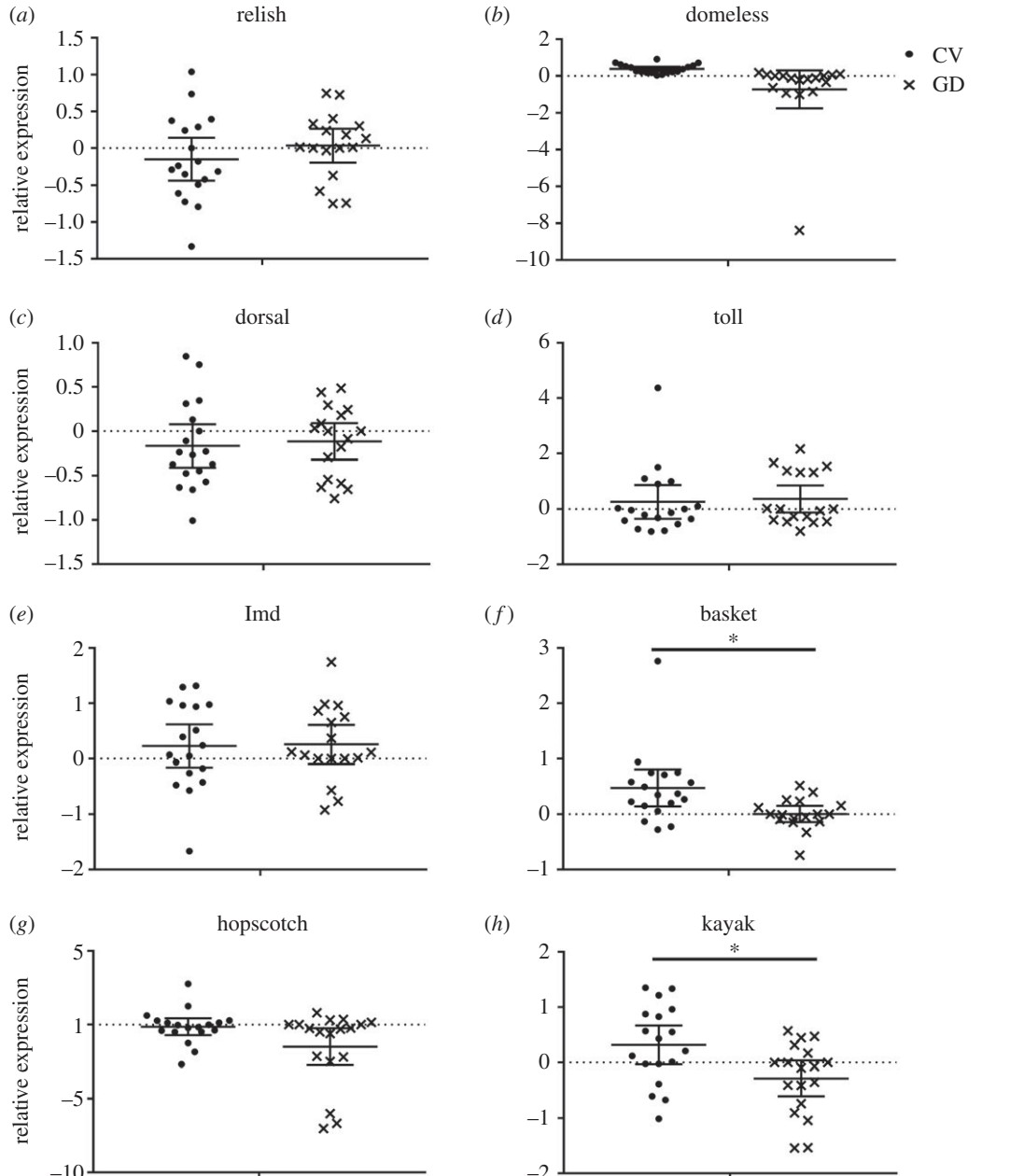

**Figure 3.** The expression level of key immune components in the abdomen of GD worker ($n = 6$ bees $\times$ 3 replicates) and CV workers ($n = 6$ bees $\times$ 3 replicates). The relative expression was $\log_2$ transformed. Means $\pm$ 95 CI are shown by the black bars and whiskers. *represents the significant difference ($p < 0.016$).

mechanical, chemical and behavioural defences as well as a complex immune system, and in addition to the host's own defences, whereby some insects are associated with protective symbionts [28]. Our study provides the first evidence of a close relation in *A. cerana* between its gut microbiota and the host immune system. The results of our study strongly emphasized the importance of commensal gut bacteria in stimulating the immune system of *A. cerana* and for strengthening the resistance of *A. cerana* to *N. ceranae* infection.

Vg, the precursor of yolk proteins, was traditionally seen as being the energy reserve for nourishment of the developing embryos, yet, its role extends beyond this nutrient function [29]. In the honeybee *A. mellifera*, it was found to participate in the regulation of social organization and individual physiology [29], and it has been linked to host immune defence as an immune-relevant molecule [30]. Thus, Vg has become widely accepted as a marker of honeybees' overall health. Accordingly, the upregulation of *Vg* expression by gut microbiota we found here clearly shows that these *A. cerana* gut inhabitants have a positive impact on host health.

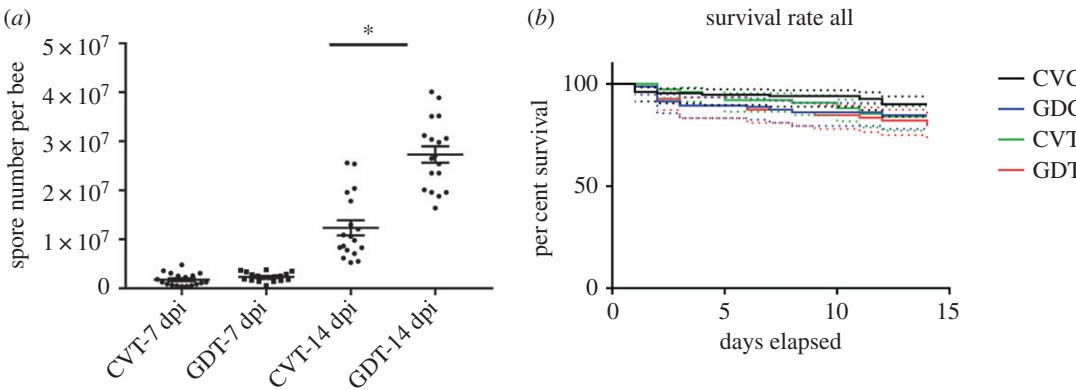

**Figure 4.** The *Nosema ceranae* spore number (*a*) in CVT and GDT worker (*n* = 6 bees × 3 replicates) after 7 dpi and 14 dpi and survival rate of *A. cerana* workers in three replicates (*b*). Means ± SEM are shown by the black bars and whiskers. * represents the significant difference (*p* < 0.05).

AMPs are crucial effectors of insects' innate immune system, providing the first line of defence against a variety of pathogens [31]. Due to their importance for maintaining honeybee health, AMPs have increasingly become a focus of research investigations. Only a reduced number of AMPs were detected in the honeybee haemolymph [32], including apidaecin, abaecin, defensin 1, defensin 2 and hymenoptaecin and their respective genes' expression is generally regulated by four intracellular signalling pathways (Toll, IMD, JNK and JAK/STAT). Many studies have shown that symbiotic bacteria can activate the immune system of their insect host and thereby increase the efficiency of pathogen defence [33,34]. Of all the AMPs expressed in honeybee haemolymph, we found that gut bacteria colonization led to significant upregulation in the expression levels of *apidaecin*, *abaecin* and *hymenoptaecin* in workers' abdomens, while the bacteria inoculation had no apparent influence on *defensin 1* and *defensin 2*. Intriguingly, Kwong *et al.* [10] found that *A. mellifera* gut microbiota can increase the expression of *apidaecin* and *hymenoptaecin*, but they had no impact on *abaecin*, suggesting that *A. cerana* and *A. mellifera* gut microbiota have similar albeit not identical functioning in host immune modulation, even though they both harbour a small, recurring set of bacterial phylotypes [6]. Experiments on mono-colonized *A. mellifera* bees have reported that *S. alvi* colonization can only upregulate *apidaecin* expression [10], whereas *F. perrara* colonization can upregulate the expression of *apidaecin*, *abaecin* and *defensin 1* [11]. Hence, host immune modulation is the outcome of complex host–microbiota interactions involving multiple phylotypes, as differing bacteria phylotypes/strains generate different microbe-associated molecular patterns for the pathway leading to AMP production. Therefore, we speculated that the difference in the regulation of AMP expression between our study and previous studies is likely to be caused by the presence of host-specific strains or the variation in the relative abundance of bacterial species in honeybees' gut microbial communities.

Recent studies have led to important advances in our understanding of the regulation of honeybee AMPs; however, there are still wide gaps between our current knowledge and the full understanding of the underlying molecular mechanisms [32]. Four non-autonomous pathways are implicated in inducible host defence, the Toll, IMD, JAK/STAT and JNK pathways, and are considered as the major directors of this process [35]. In our study, the expression of key components of these pathways was analysed. We found that key regulators of JNK pathways (*basket* and *kayak*) were significantly regulated by the colonization of gut bacteria, while the JAK/STAT, IMD and Toll pathways remained mostly unchanged. Functional study of the interaction between social bee AMP and JNK pathways is quite limited; however, these relationships have been well studied on other insects like the model *Drosophila melanogaster*. The JNK pathway has been identified as a regulator of AMP gene expression in *Drosophila* S2 cells [36] and the expression of *Drosophila* AMPs was found blocked by an inhibitor of JNK signalling and also in JNK mutant clones [37]. Given those findings, our results indicated that *A. cerana* gut symbionts enhanced host immunity though regulating JNK pathways. A study of gut microbiota dysbiosis in *A. mellifera* workers revealed that antibiotic treatment decreased the expression of *relish* [38], a transcription factor in the IMD pathway, leading to the downregulation of AMPs [12]. However, considering that gut microbiota dysbiosis and being gut microbiota deficient are different situations, it is premature to say that *A. cerana* and *A. mellifera* gut microbiota regulate different immune pathways. Therefore, studies on honeybee immune–microbiota interactions are urgently

needed, especially those that use the mono-colonized worker model or studies which combine germ-free workers and RNAi knockdown.

Colonization of the gut with commensal or mutualistic microbial communities can increase the resistance of the host against parasite invasion. For example, *Hamiltonella defensa* in pea aphids (*Acyrthosiphon pisum*) and black bean aphids (*Aphis fabae*) confer protection against parasitic wasps [39]. In addition, a variety of studies on the bumblebee showed that its intestinal symbiont of bumblebee (*Bombus terrestris*) influences the infection of the parasite *Crithidia bombi* [40,41]. Currently, *N. ceranae*, which jumped from the Asian to the western honeybee some decades ago, is the most widespread and deadly gut parasite of *A. mellifera* [15,42]. Rubanov *et al.* [43] have found that two specific strains of *Gilliamella* were significantly more abundant in bees from colonies with high *Nosema* loads versus those with low *Nosema* loads, and that eliminating their gut bacteria using antibiotics made them more susceptible to *Nosema* infections [12]. That work suggested a clear association between *A. mellifera* gut microbiota and honeybee resistance to *Nosema* spores. As its original host, *A. cerana* has coevolved with *N. ceranae* for millions of years, so it is perhaps not surprising that *N. ceranae* proliferation was inhibited by *A. cerana* gut bacteria, as significantly higher spore loads were noticed in GD workers than CV workers in our study. This result demonstrated that *A. cerana* gut microbiota contributes greatly to how the host resists this parasite. The interactions between gut microbiota, gut parasites and host are complicated; the inhibition of parasites could be accomplished through their direct interaction with microbes, or due to changes to the physical gastrointestinal landscape or the immune landscape of the gut [44]. Thus, the mechanism underlying this inhibition of a potent honeybee parasite merits further study. In addition, we have noticed that *Nosema* infection had little or no impact on honeybee longevity, in contrast with a previous study of *A. cerana* and *A. mellifera* [45]. This discrepancy could be due to the modulation of the *N. ceranae* virulence that is related to a polymorphism between *N. ceranae* isolates from different geographic origins [15]. Hence, the interspecies variants of *N. ceranae* should be explicitly considered in future studies of honeybee gut microbiota–*N. ceranae* interactions.

Taken together, our experimental work demonstrated the contribution of *A. cerana* gut microbiota to host health, pointing out the beneficial role of a balanced gut microbiome in honeybee *A. cerana* and providing new insights into the honeybee host–microbiome interactions. Also, our findings further suggest the potential use of *A. cerana* gut bacteria as probiotics for promoting honeybee resistance to *Nosema* in apiculture.

Ethics. Our study does not present research with ethical considerations.

Data accessibility. All the data generated and used in our study are included in the electronic supplementary material: table S1 includes all the gene expression data, table S2 includes the bacteria load, table S3 includes the *Nosema* spore number and table S4 includes the survival rate of *Nosema* treated CV and GD workers.

Authors' contributions. Y.W., Y.Z., H.Z. and F.H. conceived and designed the study. Y.W., Y.Z., Y.C and G.C. performed experiments. Y.W and Y.Z. analysed data and drafted the manuscript. All authors edited and approved the final version of the manuscript.

Competing interests. The authors declare no competing interests.

Funding. The work was supported by National Natural Science Foundation of China (31902222, Y.W., 31672498, H.Z. and 31872431, F.H.), Science and Technology Department of Zhejiang Province, China (2016C02054-11, F.H.) and the Modern Agroindustry Technology Research System (CARS-44, F.H. and H.Z.).

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
