## [Reviewer comments · Royal Society Open Science]

Review History

RSOS-192100.R0 (Original submission)

Review form: Reviewer 1

Is the manuscript scientifically sound in its present form?

Yes

Are the interpretations and conclusions justified by the results?

Yes

Is the language acceptable?

Yes

Do you have any ethical concerns with this paper?

No

Have you any concerns about statistical analyses in this paper?

No

Recommendation?

Accept with minor revision (please list in comments)

Comments to the Author(s)

Review on MS "Apis cerana gut microbiota contribute to host health though stimulating host immune system and strengthening host resistance to *Nosema ceranae*" by Wu, Yuqet al

In this MS the authors quantified differences in immune gene regulation and *Nosema* resistance for the honey bee species *A. cerana* with respect to reduction of a natural microbiome. They found strong differences in a variety of gene expressions and *Nosema* tolerance. This is an interesting finding and mostly confirms what is known for the honey bee *A. mellifera*. The manuscript is well written, some parts could be adapted or expanded, see my suggestions below. I would like to see some more conservative term than "germ free", because there were still moderately dense loads of bacteria in the so called germ-free samples. Other than that I only have minor, textual comments and recommend publication.

Major:

I 183: "cells per gut of CV bees and a total bacterial load of around 10^6 cells per gut of GF bees" I would not call the GF bees to be germ-free. This is a defined term. The authors should consider rephrasing this throughout the manuscript in as "underdeveloped microbiome" or similar. (also particularly in the discussion, e.g. "gut microbiota dysbiosis and germ-free are different situation" I 270)

I. 186: "homogenized guts were successfully colonized with the characteristic microbiota." This means it has the same number of bacteria, but not necessarily the same identities/ proportions of the community. As the authors didnt test for composition, this should be discussed.

Minor:

whole MS: Sometimes it is confusing, when the authors use "honey bees", it is unclear (at points used arbitrarily) whether they refer to *A. cerana*, *A. mellifera* or the whole genus. This is particularly in the introduction and discussion, and clearing that out would make it easier to catch the novelities found here.

L. 43 Please consider this article and maybe rephrase :
<https://academic.oup.com/femsle/article/366/10/fnz117/5499024>

I 115: "For GF and CV bees, six workers from each cage" Could the sampling design be explained a bit more, how many cages, same origin, same queen, how many bees in cage...?

I 118: "immediately dissected in RNase-free water. Then sampled guts were immediately placed in chilled vials separately and used for DNA extraction" DNA/RNA typo?

I. 159 "suspension containing 1×10^5 *Nosema* spores" it might be good to add a reference that this is a realistic load of *Nosema* observed also in the field.

Discussion: In some cases the study references and discusses microbiomes from very different insects, while other bees than honey bees are completely ignored. It might be good to set this also in context of other social bees (bumble bees, stingless) or even solitary bees.

Review form: Reviewer 2

Is the manuscript scientifically sound in its present form?

Yes

Are the interpretations and conclusions justified by the results?

Yes

Is the language acceptable?

No

Do you have any ethical concerns with this paper?

No

Have you any concerns about statistical analyses in this paper?

Yes

Recommendation?

Major revision is needed (please make suggestions in comments)

Comments to the Author(s)

I think that this is an interesting study, especially given that relatively little microbiome work has been conducted with *A. cerana*, the bee species which *N. ceranae* is most closely associated. I think that the experiments seem well designed, though I have some concerns about replication and sample sizes, which are unclear in this manuscript.

- 1) The description of the same sizes is not adequate. How many colonies and how many cages were used? I suggest adding a table to show the sampling. In addition, it seems that the gut bacteria for all experiments obtained from just one colony? If so, this should be justified
- 2) The statistics are also not adequately reported. Only p-values are given without the detailed statistical test results (test statistic, degrees of freedom, etc.) These should be provided.
- 3) The language throughout should be thoroughly revised since there are multiple grammatical errors. I suggest that the authors consider using an online English editing service.

Decision letter (RSOS-192100.R0)

06-Apr-2020

Dear Dr Wu,

The editors assigned to your paper ("*Apis cerana* gut microbiota contribute to host health through stimulating host immune system and strengthening host resistance to *Nosema ceranae*") have now received comments from reviewers. We would like you to revise your paper in accordance with the referee and Associate Editor suggestions which can be found below (not including confidential reports to the Editor). Please note this decision does not guarantee eventual acceptance.

Please submit a copy of your revised paper before 29-Apr-2020. Please note that the revision deadline will expire at 00.00am on this date. If we do not hear from you within this time then it will be assumed that the paper has been withdrawn. In exceptional circumstances, extensions may be possible if agreed with the Editorial Office in advance. We do not allow multiple rounds of revision so we urge you to make every effort to fully address all of the comments at this stage. If deemed necessary by the Editors, your manuscript will be sent back to one or more of the original reviewers for assessment. If the original reviewers are not available, we may invite new reviewers.

- Data accessibility

If you wish to submit your supporting data or code to Dryad (<http://datadryad.org/>), or modify your current submission to dryad, please use the following link:
<http://datadryad.org/submit?journalID=RSOS&manu=RSOS-192100>

- Competing interests

- Authors' contributions

AB carried out the molecular lab work, participated in data analysis, carried out sequence alignments, participated in the design of the study and drafted the manuscript; CD carried out the statistical analyses; EF collected field data; GH conceived of the study, designed the study,

coordinated the study and helped draft the manuscript. All authors gave final approval for publication.

- Acknowledgements

- Funding statement

Kind regards,
Lianne Parkhouse
Royal Society Open Science
openscience@royalsociety.org

on behalf of Dr Ulas Tezel (Associate Editor) and Kevin Padian (Subject Editor)
openscience@royalsociety.org

Editorial Comments to Author:

As you have been requested to edit the written English, you must provide proof that you have done so: acceptable proof includes a certificate of language-editing from a language editing service or a signed letter from a native speaker of English. If you do not provide this proof, your manuscript may be returned to you.

For information about language editing services endorsed by the Royal Society, please follow the link below:

<https://royalsociety.org/journals/authors/language-polishing/>

Reviewers' Comments to Author:

Reviewer: 1

Comments to the Author(s)

Review on MS "Apis cerana gut microbiota contribute to host health through stimulating host immune system and strengthening host resistance to *Nosema ceranae*" by Wu, Yuqet al

In this MS the authors quantified differences in immune gene regulation and *Nosema* resistance for the honey bee species *A. cerana* with respect to reduction of a natural microbiome. They found strong differences in a variety of gene expressions and *Nosema* tolerance. This is an interesting finding and mostly confirms what is known for the honey bee *A. mellifera*. The manuscript is well written, some parts could be adapted or expanded, see my suggestions below. I would like to see some more conservative term than "germ free", because there were still moderately dense loads of bacteria in the so called germ-free samples. Other than that I only have minor, textual comments and recommend publication.

Major:

I 183: "cells per gut of CV bees and a total bacterial load of around 10^6 cells per gut of GF bees" I would not call the GF bees to be germ-free. This is a defined term. The authors should consider rephrasing this throughout the manuscript in as "underdeveloped microbiome" or similar. (also

particularly in the discussion, e.g. "gut microbiota dysbiosis and germ-free are different situation" (L 270)

L. 186: "homogenized guts were successfully colonized with the characteristic microbiota." This means it has the same number of bacteria, but not necessarily the same identities/proportions of the community. As the authors didn't test for composition, this should be discussed.

Minor:

whole MS: Sometimes it is confusing, when the authors use "honey bees", it is unclear (at points used arbitrarily) whether they refer to *A. cerana*, *A. mellifera* or the whole genus. This is particularly in the introduction and discussion, and clearing that out would make it easier to catch the novelities found here.

L. 43 Please consider this article and maybe rephrase :
<https://academic.oup.com/femsle/article/366/10/fnz117/5499024>

L 115: "For GF and CV bees, six workers from each cage" Could the sampling design be explained a bit more, how many cages, same origin, same queen, how many bees in cage...?

L 118: "immediately dissected in RNase-free water. Then sampled guts were immediately placed in chilled vials separately and used for DNA extraction" DNA/RNA typo?

L. 159 "suspension containing 1×10^5 *Nosema* spores" it might be good to add a reference that this is a realistic load of *Nosema* observed also in the field.

Discussion: In some cases the study references and discusses microbiomes from very different insects, while other bees than honey bees are completely ignored. It might be good to set this also in context of other social bees (bumble bees, stingless) or even solitary bees.

Reviewer: 2

Comments to the Author(s)

I think that this is an interesting study, especially given that relatively little microbiome work has been conducted with *A. cerana*, the bee species which *N. ceranae* is most closely associated. I think that the experiments seem well designed, though I have some concerns about replication and sample sizes, which are unclear in this manuscript.

1) The description of the same sizes is not adequate. How many colonies and how many cages were used? I suggest adding a table to show the sampling. In addition, it seems that the gut bacteria for all experiments obtained from just one colony? If so, this should be justified

2) The statistics are also not adequately reported. Only p-values are given without the detailed statistical test results (test statistic, degrees of freedom, etc.) These should be provided.

3) The language throughout should be thoroughly revised since there are multiple grammatical errors. I suggest that the authors consider using an online English editing service.

Author's Response to Decision Letter for (RSOS-192100.R0)

See Appendix A.

Decision letter (RSOS-192100.R1)

Dear Dr Wu,

It is a pleasure to accept your manuscript entitled "Apis cerana gut microbiota contribute to host health through stimulating host immune system and strengthening host resistance to *Nosema ceranae*" in its current form for publication in Royal Society Open Science. The comments of the reviewer(s) who reviewed your manuscript are included at the foot of this letter.

on behalf of Dr Ulas Tezel (Associate Editor) and Kevin Padian (Subject Editor)
openscience@royalsociety.org

Appendix A

Editorial Comments to Author:

As you have been requested to edit the written English, you must provide proof that you have done so: acceptable proof includes a certificate of language-editing from a language editing service or a signed letter from a native speaker of English. If you do not provide this proof, your manuscript may be returned to you.

For information about language editing services endorsed by the Royal Society, please follow the link below:

<https://royalsociety.org/journals/authors/language-polishing/>

Reply: We have had our manuscript edited for correct English language usage, grammar, punctuation and spelling by qualified native English speaking editors at Charlesworth Author Services.

Reviewers' Comments to Author:

Reviewer: 1

Comments to the Author(s)

Review on MS "Apis cerana gut microbiota contribute to host health though stimulating host immune system and strengthening host resistance to *Nosema ceranae*" by Wu, Yuqet al

In this MS the authors quantified differences in immune gene regulation and *Nosema* resistance for the honey bee species *A. cerana* with respect to reduction of a natural microbiome. They found strong differences in a variety of gene expressions and *Nosema* tolerance. This is an interesting finding and mostly confirms what is known for the honey bee *A. mellifera*. The manuscript is well written, some parts could be adapted or expanded, see my suggestions below. I would like to see some more conservative term than "germ free", because there were still moderately dense loads of bacteria in the so called germ-free samples. Other than that I only have minor, textual comments and recommend publication.

Reply: We thank the reviewer for the compliments on our study. We have revised and improved our manuscript according to the reviewer's comments.

Major:

l 183: "cells per gut of CV bees and a total bacterial load of around 10^6 cells per gut of GF bees" I would not call the GF bees to be germ-free. This is a defined term. The authors should consider rephrasing this throughout the manuscript in as "underdeveloped microbiome" or similar. (also particularly in the discussion, e.g. "gut microbiota dysbiosis and germ-free are different situation" l 270)

Reply: We appreciate the suggestion from reviewer and have rephased germ-free worker as Gut microbiota deficient (GD).

l. 186: "homogenized guts were successfully colonized with the characteristic microbiota." This means it has the same number of bacteria, but not necessarily the same identities/proportions of the community. As the authors didnt test for composition, this should be discussed.

Reply: While we didn't test the gut community composition of our samples, the method used is widely used in studies of honey bee gut microbiota. And we do have tested the gut community in another study, which is based on *A. mellifera*, the result showed that workers were colonized with the characteristic microbiota. So, we believe our method is appropriate for our study and reliable. However, we agree that it is not very accurate to state "successfully colonized with the characteristic microbiota", so we have rephrased our sentence.

The top 10 bacterial genus identified in the gut of CV and GD workers using 16s rRNA sequencing.

Minor:

whole MS: Sometimes it is confusing, when the authors use "honey bees", it is unclear (at points used arbitrarily) whether they refer to *A. cerana*, *A. mellifera* or the whole genus. This is particularly in the introduction and discussion, and clearing that out would make it easier to catch the novelities found here.

Reply: We apology for our confusing usage of "honey bees" and have changed it to specific bee species in our manuscript.

L. 43 Please consider this article and maybe rephrase : <https://academic.oup.com/femsle/article/366/10/fnz117/5499024>

Reply: It is interesting to know the existence of "microbiome-free" animals, so we have rephrased the sentence to "some of which are emerging as key players in governing host health".

l 115: "For GF and CV bees, six workers from each cage" Could the sampling design be explained a bit more, how many cages, same origin, same queen, how many bees in cage...?

Reply: In order to provide a better presentation of our experiment design, we have re drafted the Material and methods section. A total of 3 *A. cerana* colonies were used and workers from the same colony/queen in each replicate were used. The number of bees per cage and the total number of cages were added. The sampling timepoints were also included.

l 118: "immediately dissected in RNase-free water. Then sampled guts were immediately placed in chilled vials separately and used for DNA extraction" DNA/RNA typo?

Reply: We are sorry for the mistake here and have corrected this mistake.

l. 159 "suspension containing 1×10^5 Nosema spores" it might be good to add a reference that this is a realistic load of Nosema observed also in the field.

Reply: The load of *Nosema* spore used here is a commonly used load for laboratory inoculation, and we have added a relative reference.

Discussion: In some cases the study references and discusses microbiomes from very different insects, while other bees than honey bees are completely ignored. It might be good to set this also in context of other social bees (bumble bees, stingless) or even solitary bees.

Reply: To provide more information, we have added several studies about bumble bee gut bacteria and host health.

Reviewer: 2

Comments to the Author(s)

I think that this is an interesting study, especially given that relatively little microbiome work has been conducted with *A. cerana*, the bee species which *N. ceranae* is most closely associated. I think that the experiments seem well designed, though I have some concerns about replication and sample sizes, which are unclear in this manuscript.

Reply: We appreciate the feedbacks from the reviewer. We have revised and improved our manuscript according to the reviewer's comments.

1) The description of the same sizes is not adequate. How many colonies and how many cages were used? I suggest adding a table to show the sampling. In addition, it seems that the gut bacteria for all experiments obtained from just one colony? If so, this should be justified

Reply: We are sorry for our inadequate description of our sample size and have redrafted the Material and methods section. The gut bacteria used were obtained from three colonies, as three different *A. cerana* colonies were used in our study.

2) The statistics are also not adequately reported. Only p-values are given without the detailed statistical test results (test statistic, degrees of freedom, etc.) These should be provided.

Reply: We have provided a detailed statistical test results including the t statistic, degrees of freedom and p-value.

3) The language throughout should be thoroughly revised since there are multiple grammatical errors. I suggest that the authors consider using an online English editing service.

Reply: We apology for our poor English writing, the revised manuscript was edited by language editing service.